# Quantifying cumulative phenotypic and genomic evidence for procedural generation of metabolic network reconstructions

Thomas J. Moutinho, Jr. [ID], Benjamin C. Neubert [ID], Matthew L. Jenior [ID], Jason A. Papin [ID] *

University of Virginia, Department of Biomedical Engineering, Charlottesville, Virginia, United States of America

* papin@virginia.edu

## Abstract

Genome-scale metabolic network reconstructions (GENREs) are valuable tools for understanding microbial metabolism. The process of automatically generating GENREs includes identifying metabolic reactions supported by sufficient genomic evidence to generate a draft metabolic network. The draft GENRE is then gapfilled with additional reactions in order to recapitulate specific growth phenotypes as indicated with associated experimental data. Previous methods have implemented absolute mapping thresholds for the reactions automatically included in draft GENREs; however, there is growing evidence that integrating annotation evidence in a continuous form can improve model accuracy. There is a need for flexibility in the structure of GENREs to better account for uncertainty in biological data, unknown regulatory mechanisms, and context-specificity associated with data inputs. To address this issue, we present a novel method that provides a framework for quantifying combined genomic, biochemical, and phenotypic evidence for each biochemical reaction during automated GENRE construction. Our method, Constraint-based Analysis Yielding reaction Usage across metabolic Networks (CANYUNs), generates accurate GENREs with a quantitative metric for the cumulative evidence for each reaction included in the network. The structuring of CANYUNs allows for the simultaneous integration of three data inputs while maintaining all supporting evidence for biochemical reactions that may be active in an organism. CANYUNs is designed to maximize the utility of experimental and annotation datasets and to ultimately assist in the curation of the reference datasets used for the automatic construction of metabolic networks. We validated CANYUNs by generating an *E. coli* K-12 model and compared it to the manually curated reconstruction iML1515. Finally, we demonstrated the use of CANYUNs to build a model by generating an *E. coli* Nissle CANYUNs model using novel phenotypic data that we collected. This method may address key challenges for the procedural construction of metabolic networks by leveraging uncertainty and redundancy in biological data.

**Data Availability Statement:** All code and data used is available on Github at github.com/Tjmoutinho/CANYUNs.

**Funding:** This work was funded by the National Institutes of Health (Grant number: AT010253; Authors funded: JP, TM, BN, MJ). Additionally, the University of Virginia, School of Medicine, Sture G. Olsson Graduate Fellowship and the National Science Foundation Graduate Research Fellowship Program funded TM. The funders had no role in study design, data collection and analysis, decision to publish, or preparation of the manuscript.

**Competing interests:** The authors have declared that no competing interests exist.

## Author summary

Genome-scale metabolic network reconstructions (GENREs) are a mathematical representation of the biochemical reactions that may occur in an organism. GENREs are built using known biochemistry, organism-specific genetics, and other types of context-specific data. There is a need for automated GENRE construction methods that better quantify uncertainty in biological data and context-specific data inputs. To address this issue, we present a novel method that provides a framework for quantifying combined genetic, biochemical, and growth assay data for each biochemical reaction during the automated GENRE construction process. Our method, Constraint-based Analysis Yielding reaction Usage across metabolic Networks (CANYUNs), generates accurate GENREs with a quantitative metric for the cumulative evidence that is associated with each reaction in the network. CANYUNs is designed to maximize the utility of data inputs and to assist in the curation of the reference datasets used for the automatic construction of GENREs. We validated CANYUNs by building an *E. coli* K-12 model and compared it to the manually curated reconstruction iML1515. Finally, we demonstrated the use of CANYUNs to build a model by building an *E. coli* Nissle CANYUNs model using novel growth assay data that we collected.

## Introduction

Complex microbial communities play an important role in human physiology and environmental processes [1–7]. Genome-scale metabolic network reconstructions (GENREs) have been shown to model the functional capabilities of microbes and their interactions in communities [8–11]. A GENRE is a constraint-based model structure that enables the combination of various forms of biological data to gain an improved mechanistic understanding of metabolism [12]. This form of modeling explicitly accounts for biochemical thermodynamics and stoichiometry to represent the physical constraints that govern cellular metabolism. Methods used to generate GENREs are progressively being automated to reduce time and resource requirements with the goal of modeling the vast number of unique species and strains that reside in human-associated microbiota [13–16]. However, there remains a need for advancements in the procedural generation of GENREs to improve the accounting of uncertainty in the source biological data.

The foundational data that procedurally generated GENREs are built upon is a universal biochemical reaction network with associated reference genetic annotation data for sequence-to-reaction mapping. When building an organism-specific GENRE, a genome is annotated with precise biochemical reactions. The annotation process typically involves a threshold of sequence alignment that is used to determine if a sequence is similar enough to the reference sequence to justify annotation with the associated biochemical function [17]. The data used to build GENREs is incomplete and subject to uncertainty, necessitating gapfilling of the metabolic network generated via genetic data alone. Gapfilling is the process of adding biochemical reactions with low or no genetic evidence to a GENRE based on functional phenotypic growth data and cellular biomass synthesis requirements. The resulting accuracy of the curated GENRE is then calculated by how well it recapitulates the phenotypic growth data utilized for training. There are additional methods for further assessment of model quality involving other data types, such as gene essentiality data, and separate validation data [18]. Recent methods have demonstrated that utilizing gene annotation alignment scores in a continuous way can help to improve gapfilling results [14,19].

A curated GENRE consists of a set of reactions that have biological evidence suggesting that they are catalyzed by the organism. When a GENRE is procedurally generated, the remaining error is commonly dominated by false growth calls; these errors indicate that these models

over-predict the metabolic capabilities of an organism. Additionally, annotation alignment scores, universal biochemical network source data, and annotation reference data are all often left out from published GENREs [20,21]. Without the source sequence-to-reaction data used to generate a GENRE, the reactions that are included in the curated GENRE lack explicit indication of what type of biological evidence was used to justify inclusion.

Phenotypic growth data used for gapfilling is not utilized as context specific data to contextualize a GENRE. Growth data is typically used during gapfilling without accounting for the additional regulatory machinery that may be acting on the metabolic system. Whereas, transcriptomics data can be used to gain context specific insight into how metabolic flux might be routed in an unknown growth condition [22–24]. We hypothesize that phenotypic growth data provides a similar type of context-specific data compared to transcriptomic data, yet it is utilized to build a metabolic network rather than contextualize the existing universal metabolic network. From a biological point of view, it is an over simplification of an organism-specific metabolic network to ignore the existence of gene regulation, that suppresses gene expression, during the GENRE building process [25]. The practice of including all genetic data in the functional GENRE and then gapfilling remaining essential reactions with phenotypic data results in an over-constrained assessment of the biological system under investigation [14]. There is a need for additional flexibility in the structure of GENREs to better account for uncertainty in biological data, unknown regulatory mechanisms, and context-specificity associated with phenotypic growth data inputs.

In this study, we present a novel method for contextualizing a manually curated universal metabolic network through the simultaneous integration of genetic annotation data and phenotypic growth data. Our method, Constraint-based Analysis Yielding reaction Usage across metabolic Networks (CANYUNs), procedurally generates a GENRE by explicitly quantifying the combined biological evidence for the inclusion of reactions in the resulting network. CANYUNs utilizes a continuous weighting for each reaction in a curated universal metabolic network to quantify the evidence provided by the biological data that is used during the reconstruction process. Rather than gapfilling a draft network by leveraging phenotypic data, CANYUNs determines the reactions required for computational growth in each known growth condition separately to quantify the cumulative evidence for each reaction. The cumulative evidence generated for each reaction during the CANYUNs training process is subsequently used to determine the reactions that are included in the final GENRE. The resulting CANYUNs model consists of the universal metabolic network and associated reference annotations, organism-specific genetic alignment scores, phenotypic growth data, and certainty values associated with each reaction included in the curated network.

## Results

### Constraint-based Analysis Yielding reaction Usage across metabolic Networks (CANYUNs)

The model training process in CANYUNs is designed to capture and quantify the cumulative experimental and genomic evidence for the inclusion of biochemical reactions in a procedurally generated GENRE. CANYUNs simultaneously utilizes phenotypic growth data, genomic annotation evidence, and universal biochemical network data making it distinct from existing reconstruction methods that first reconstruct a draft metabolic network using genomic data and then sequentially gapfill additional reactions to match model predictions with phenotypic experimental data. CANYUNs maintains a direct connection with all annotation evidence used during model building to help facilitate future model curation.

We built a curated universal biochemical network by combining the reactions from the CarveMe universal network [14] and reactions from the manually curated *E. coli* K-12 model,

iML1515 [26]. When metabolite formulas did not match, we used the iML1515 formulas to maximize the number of mass-balanced reactions in the final universal network. For additional curation, we used an optimization method to check the network for generation of free-mass (see Methods). In short, we created intracellular sink reactions for each intracellular metabolite in the network and closed all exchange reactions to ensure the network did not have access to any extracellular metabolites. We then maximized the sum of flux through all sink reactions to identify any metabolites produced due to mass-imbalanced reactions or mass-generating loops. We curated the universal network by manually removing reactions that were contributing to free-mass generation. Using this optimization-based method, we were able to more rigorously identify free-mass generation in the network compared to simply checking each reaction for mass-balance. The universal metabolic network and a list of reactions removed are available on GitHub (see Methods).

We utilized BLASTp to align the genome of the target organism with reference sequences in the CarveMe sequence-to-reaction dataset. We used the sequence alignment bitscores for *E. coli* K-12 genes and the CarveMe dataset to then generate reaction bitscores using the published method [14]. We subsequently used a step-wise linear transformation to convert the reaction bitscores to reaction weights that fall between -1 and 1 to use during linear optimization and flux balance analysis. We developed a novel formulation of flux balance analysis called, Data Guided Flux Balance Analysis (dgFBA) specifically for CANYUNs. This optimization equation minimizes the sum of flux through all reactions with low or no genetic evidence while simultaneously maximizing the sum of flux through all reactions with substantial genetic evidence. The degree to which a reaction is minimized or maximized is linearly determined by the reaction weights. During a dgFBA optimization, flux is required through the biomass reaction to represent growth. Importantly, dgFBA allowed us to determine the flux-carrying reactions (FCRs) in each experimental growth condition by setting the exchange reactions to represent the specific growth media conditions. By tracking the FCRs for each growth condition, we were able to then calculate the ratio of growth conditions in which a reaction carries flux and determine reaction Certainty Values (CVs) for each FCR indicating confidence in the presence of each biochemical function in the target organism.

In the final stage, all flux-carrying reactions across the experimental growth conditions are used to generate an organism-specific CANYUNs model (Fig 1A). The resulting network is processed further by selectively removing a single reaction, or a small set of reactions, to further improve the overall accuracy of the model and adjust the type of error remaining. For validation of CANYUNs, we generated an automatic GENRE for *E. coli* K-12 leveraging phenotypic nutrient utilization data obtained from EcoCyc and compared it to iML1515 [26–28].

### Data guided flux balance analysis

The reaction bitscores for *E. coli* K-12 were calculated directly from BLASTp sequence alignment bitscores using a previously published method [14]. One third of reactions in the universal network have a bitscore of 0 and the rest range from 1 to 2,500 (Fig 2A). A typical bitscore threshold for assigning a reference enzymatic metabolic function to the query sequence(s) is between 200 and 500 [29–31]. The level of confidence in a functional call increases with the value of the bitscore, yet small changes in a sequence can result in large functional changes. Bitscores below the threshold also contribute information about the protein in question, values that are just shy of the threshold may still have the same function as the reference protein; however, scores that fall far short of the threshold suggest that the protein in question does not have the function of the reference.

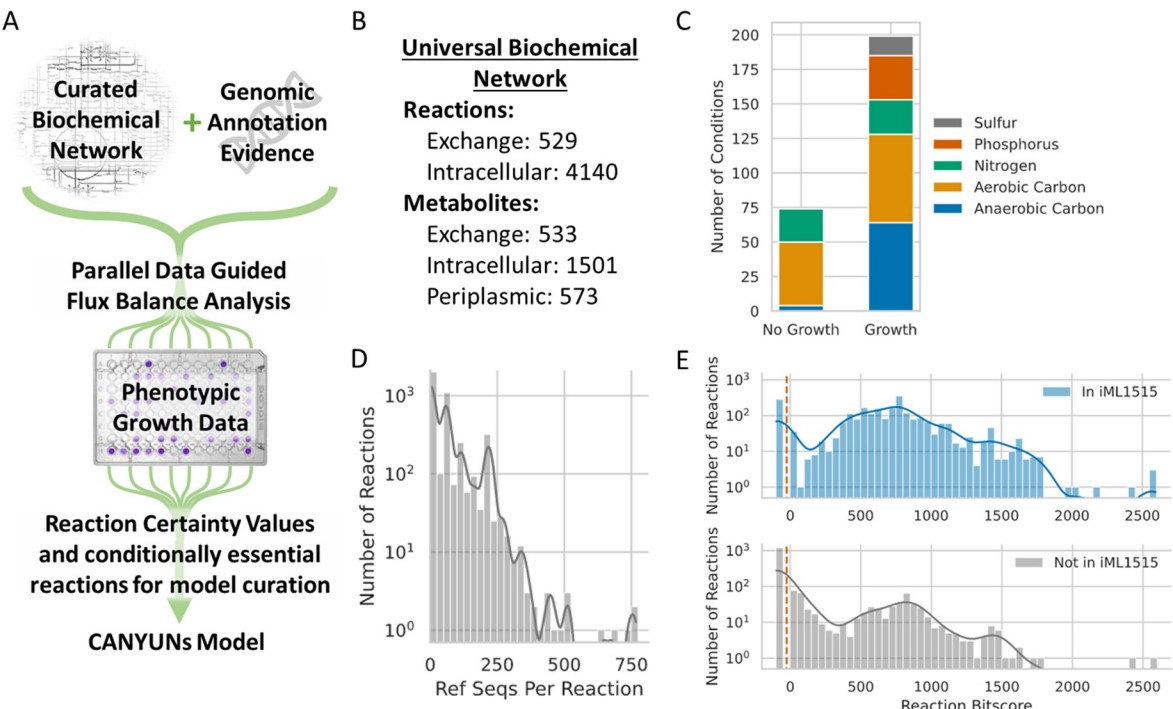

**Fig 1. The CANYUNs pipeline integrates biochemical, phenotypic, and genomic data to quantitatively identify reactions that are likely catalyzed by an organism.** (A) Genomic annotation data and phenotypic growth data for a specific organism are used to influence the flux distribution through a curated universal biochemical network to build an organism-specific metabolic network model. Parallel growth simulations using Data Guided Flux Balance Analysis for each known experimental growth condition allows for a model building process that is not influenced by the order in which growth conditions are integrated. This process allows for the explicit quantification of reaction Certainty Values, determined by the ratio of times a reaction carries flux across all of the condition-specific solutions to the total number of conditions. (B) The universal biochemical network used in this study consists of reactions from the CarveMe dataset as well as novel reactions added from the manually curated *E. coli* metabolic network, iML1515. (C) The phenotypic data used in this study includes Biolog minimal media growth data from ~275 different conditions. (D) The sequence-to-reaction dataset used to calculate reaction annotation evidence consists of over 4,000 reactions with 1 to 800 sequences associated with each reaction. (E) The distribution of reaction bitscores for *E. coli* K-12 shows that there are reactions in the universal network with high evidence that are not included in iML1515. There are also many reactions with low evidence that are not included in iML1515, as expected. The annotation evidence generated for *E. coli* K-12 shows that there are 1,460 reactions in the universal biochemical network that have no genetic evidence associated with them (left of the dashed orange line), 260 of these reactions are in iML1515 and 1,200 of them are not.

We designed dgFBA to account for some of the uncertainty inherent in setting a threshold for assigning function to a given protein by utilizing reaction weights that are a function of the reaction bitscores. The reaction weights influence the reactions that carry flux in the optimization solution. Genomic annotation data must be transformed to a range of values that are compatible with dgFBA. The transformation function used in this study is graphically displayed in Fig 2B. This function can be adjusted based on the user's preferences. In this study we selected a bitscore threshold of 500 based on a sensitivity analysis that demonstrated that model accuracy was insensitive to values between 200 and 500 (Fig 2B). The resulting reaction weights are more evenly distributed between -1 and 1 with the high bitscore reactions all receiving a weighting of 1 and the reactions without bitscores all receiving a weighting of -1 (Fig 2C and 2D).

FCRs in a dgFBA solution are a result of complex interactions among the reaction weight values, media condition, and flux demands (i.e. biomass). Flux is maximized through reactions with a bitscore above 500 and minimized through reactions below 500. However, the degree of maximization and minimization depend upon the value of the bitscore. The low evidence reactions that are included in the final flux solution are likely essential for flux through biomass

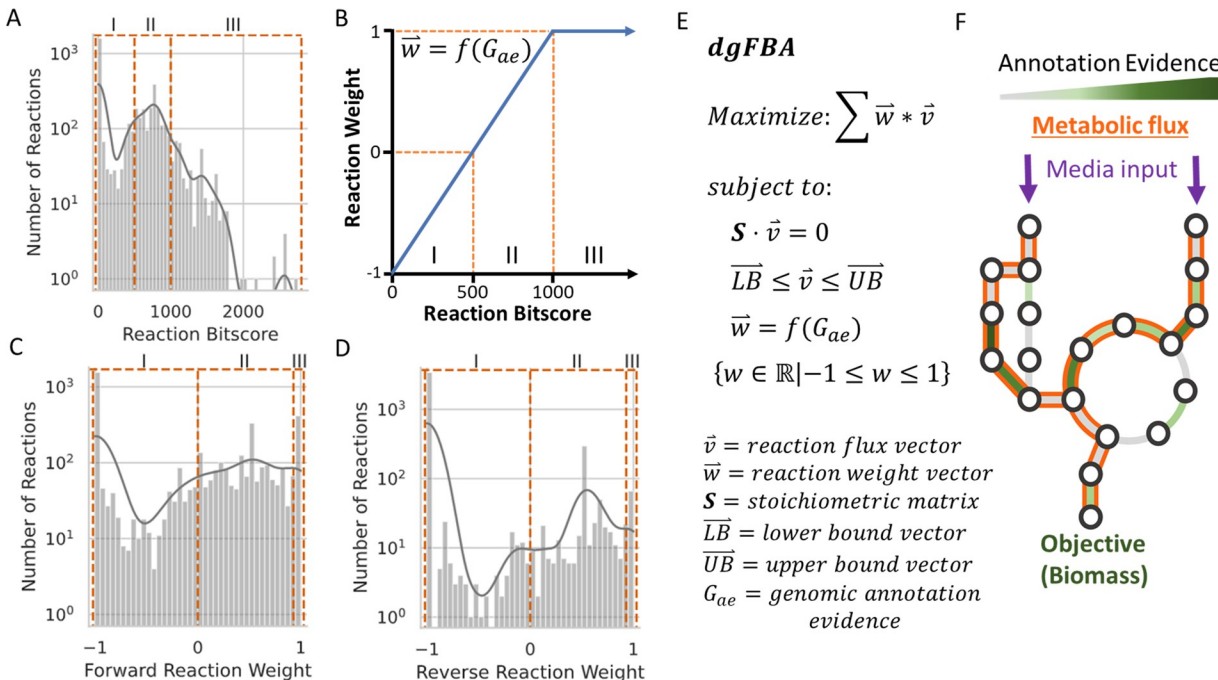

**Fig 2. Data Guided Flux Balance Analysis.** (A) Distribution of reaction bitscores for *E. coli* K-12. (B) This figure is a visual representation of the transformation function for calculating the reaction weights based on reaction bitscores. The reaction bitscore of 500 corresponds with zero in the weight space. (C) Distribution of the calculated weights for forward reactions. (D) The distribution of weights for reverse reactions shows that there are far fewer reactions that allow flux in both directions or only in the reverse direction. (E) Data Guided Flux Balance Analysis optimization problem. Reactions with a positive weight are maximized and reactions with a negative weight are minimized proportional to the value of the weight. (F) Toy network example demonstrating the flux-carrying reactions that would result from the pictured annotation evidence distribution and media inputs.

and can be thought of as gapfilled reactions that maintain their genomic annotation evidence (Fig 2E and 2F). Utilizing the reaction bitscores in this way allows for additional flexibility with reactions near the bitscore threshold of 500 where the reaction weight is equal to zero. Reactions with weights near zero are much less impacted by the dgFBA objective function and therefore are influenced far greater by thermodynamic requirements.

Data Guided Flux Balance Analysis can be compared to parsimonious enzyme usage flux balance analysis (pFBA) to demonstrate how flux through the network changes with additional layers of information. The objective of pFBA is to uniformly minimize the sum of flux across all reactions, while maintaining flux through the biomass reaction [32]. Since dgFBA maximizes the weighted flux through reactions with genetic evidence the flux distribution is consistently different from the pFBA flux distribution. However, the two optimization problems remain similar because the majority of reactions in the universal network do not have genetic evidence and are thus minimized in a dgFBA problem, just as they are in a pFBA problem. The flux distribution generated using dgFBA does not represent what the biological flux might be; it is utilized in a binary way to determine which reactions carry flux and which do not. We compared dgFBA to pFBA to quantify how much impact the genetic data has on the flux distribution for each solution and thus the difference in the set of reactions that are required to carry non-zero flux for simulated growth. We generated a separate pFBA and dgFBA flux solution for each known *E. coli* K-12 growth condition. We used two metrics to verify that dgFBA results in more FCRs, while also increasing the number of reactions that have associated annotation evidence (Fig 3A and 3B). We found that across all 199 growth conditions, the average

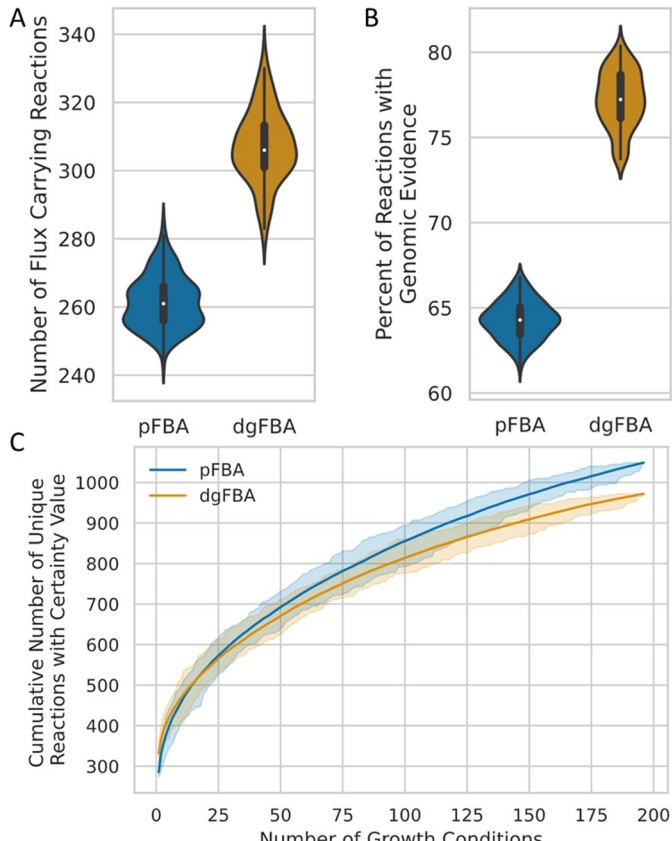

**Fig 3. Data Guided Flux Balance Analysis breaks parsimony and identifies fewer unique reactions required for simulated growth on all experimental growth conditions for _E. coli_ K-12.** (A) The number of FCRs in each growth condition is visualized for pFBA and dgFBA to quantify the degree to which dgFBA breaks parsimony. (B) The number of reactions with bitscores above 500 that carry flux in a dgFBA solution is greater than the number in a pFBA solution. (C) The cumulative number of unique FCRs identified by dgFBA is fewer than pFBA. The complete range in number of unique FCRs is indicated by the shaded regions.

number of FCRs for dgFBA was 305, which was 45 reactions greater than the average for pFBA solutions. We expected dgFBA to identify less parsimonious solutions than pFBA due to the influence imparted on the solution from the annotation evidence. It is important to note that the number of FCRs for the dgFBA solution in a single condition was always greater than the number of FCRs in the corresponding pFBA solution by at least 10 FCRs. A second important verification was to ensure that dgFBA also identifies solutions that contain a greater proportion of FCRs that have associated annotation evidence. We found that the average number of FCRs with annotation evidence greater than or equal to our reaction bitscore threshold of 500 in dgFBA solutions was nearly 20% greater than the pFBA solutions.

To determine the FCRs across all of the known growth conditions, we generated rarefaction curves each consisting of 10,000 samples to measure the full distribution of unique permutations of growth conditions that could be used to generate the GENRE. The x-axis displays the number of growth conditions used to calculate the total number of unique FCRs found (Fig 3C). The shaded regions show the minimum and maximum values sampled for each number of conditions included. The small range between the minimum and maximum indicates that there is minimal advantage to optimizing for the minimum number of growth conditions that provide the maximum training value. Each individual growth condition adds unique reactions

to the cumulative set of unique FCRs. However, the asymptotic shape of the average curves indicates that the total number of valuable unique minimal growth conditions may not be far beyond 200 conditions. The number of unique FCRs identified by dgFBA across all growth conditions is fewer than pFBA, indicating that there is a core set of FCRs with genetic evidence that dgFBA preferentially identifies over pFBA (Fig 3C). These data indicate that dgFBA performs as intended; reactions with genetic evidence preferentially carry flux even when there is a more parsimonious path which results in a diversion of flux away from extraneous reactions that are more parsimonious but lack sufficient genetic evidence.

## Certainty values determine the reactions that are included in the CANYUNs Model for *E. coli* K-12

The CANYUNs pipeline involves generating a dgFBA solution for each of the known growth conditions using the curated universal metabolic network. During the process of recording the FCRs for each condition, the directionality of each flux value is used to specifically determine the cumulative evidence for each reaction specific to direction. We calculated a reaction certainty value for each reaction in the universal metabolic network based on the set of FCRs from each growth condition. The Certainty Value (CV) for a reaction is the ratio of the number of times the reaction carries flux in the known growth conditions over the total number of known growth conditions. A CV indicates the cumulative experimental evidence for the presence of the biochemical function in an organism-specific metabolic network. Using the *E. coli* K-12 genome and phenome, we calculated 690 reactions with CVs greater than zero in the forward direction (Fig 4A), and 127 reactions with CVs greater than zero in the reverse direction (Fig 4B). There are 46 FCRs that were found to have both forward and reverse reaction CVs (Fig 4C).

We built an *E. coli* K-12 specific GENRE that consists of the reactions identified to have CVs greater than zero, including only the reaction directionalities specifically with CVs. Reversible reactions that receive a CV above zero in only one direction were set to only allow flux in that direction. Reactions that have genetic evidence, but have CVs of zero are not included in the CANYUNs model. We simulated growth in each of the known growth conditions using the resulting CANYUNs model to determine the baseline performance. The draft CANYUNs model, at this point, had an overall accuracy of 80% with a strong bias toward false positive growth calls (Fig 4D). To improve the model accuracy, we calculated the conditionally essential reactions for each of the conditions predicted to allow for growth, including false growth predictions. A comparison across the sets of conditionally essential reactions revealed reactions that, when removed, would provide a net benefit for improving the overall accuracy of the adjusted model. We identified that with the removal of a single reaction, RuBisCO, the number of false positives was reduced by 38 conditions and the number of true positives was only decreased by 7 conditions. RuBisCO was manually selected for removal because it had the maximum net benefit of 31 conditions and the least annotation evidence. All of the other candidate reactions for removal are plotted based on their net benefit to accuracy upon removal versus their annotation evidence value (Fig 4E). This process could be repeated for further alteration of the model. Although RuBisCO is an obvious reaction that should not have been included in the universal metabolic network for microbial systems, due to its involvement with photosynthesis; this result demonstrates that there are reactions that may require manual removal from the universal metabolic network based on additional biological knowledge aside from a lack of annotation evidence or contribution to mass-generating loops. However,

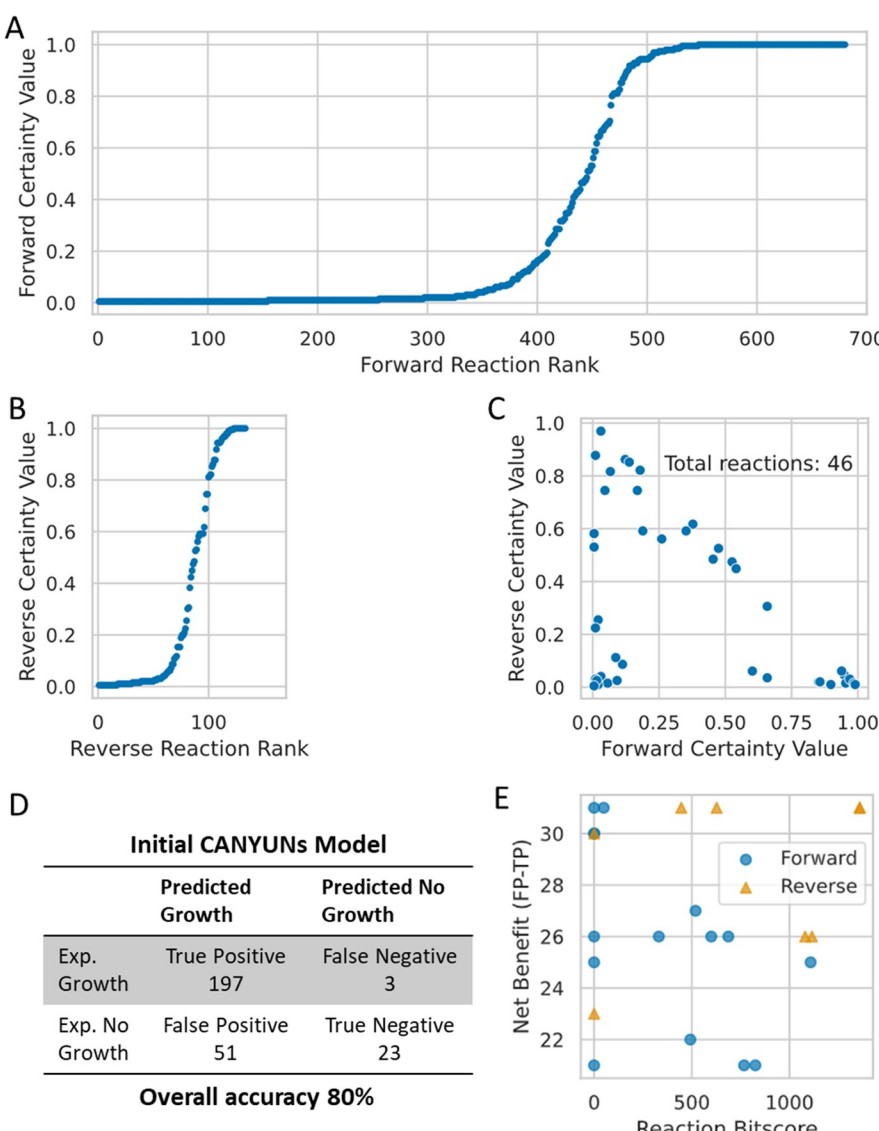

**Fig 4. *E. coli* K-12 CANYUNs model generation and draft processing.** (A) Ranked scatter plot of forward reaction Certainty Values. (B) Reverse reaction Certainty Values. (C) Certainty values for reversible reactions that carry flux in both directions. (D) Initial accuracy of CANYUNs model before curation of the universal biochemical network is 80% with a Matthews Correlation Coefficient of 0.45. (E) Simulation of conditionally essential reactions allow for the user to identify reactions that can be selectively removed from the resulting model that improve the overall predictive accuracy. The net benefit refers to the number of false positives that will be corrected minus the number of true positives lost due to removing a given reaction. RuBisCO is the forward reaction in the top left corner of the plot with maximum net benefit and minimum genetic evidence.

CANYUNs allowed for rapid identification of reactions that may be improperly included during the process of procedural generation of a CANYUNs model.

## CANYUNs more accurately recapitulates phenotypic data

The final *E. coli* K-12 model from the CANYUNs pipeline can be compared with two automatically generated models using CarveMe and the manually curated model iML1515 to benchmark and validate its performance. Using the same input biochemical and genetic data as the

CANYUNs model, we generated a CarveMe model without using phenotypic data to establish how subsequent gapfilling impacts the model accuracy (Fig 5A). The gapfilled CarveMe model that we generated had an overall accuracy of 76%, a 24% improvement over the untrained model (Fig 5B). The training process results in nearly all of the false negative predictions being corrected, as can be expected. The manually curated reconstruction, iML1515, was not specifically curated for all of the known growth conditions used to train the CarveMe model and the CANYUNs model, but it remains a valuable point of comparison as the best representation of *E. coli* K-12 metabolism that is currently available. Our *E. coli* K-12 CANYUNs model shows the highest overall accuracy, while maintaining a balance in type 1 and type 2 error. The distinction between false positives and false negatives is notable because false negatives represent an opportunity to selectively add organism-specific reactions to the universal model that directly corrects the issue. However, correcting false positive errors involves finding reactions to remove or adjust that result in minimal negative impacts to the rest of the network. CANYUNs provides a method for selectively adjusting the balance of error based on user preferences during the construction of the GENRE.

## CANYUNs more accurately identifies the reactions present in iML1515

It is possible to generate a CANYUNs model using pFBA instead of dgFBA; in this case no genetic data is incorporated to influence the flux distribution of the solution for each growth

**A** **CarveMe Model without Gapfilling**

| | Predicted Growth | Predicted No Growth |
|---|---|---|
| Exp. Growth | True Positive 100 | False Negative 96 |
| Exp. No Growth | False Positive 30 | True Negative 39 |
| | Overall accuracy 52% | |

**B** **CarveMe Gapfilled Model**

| | Predicted Growth | Predicted No Growth |
|---|---|---|
| Exp. Growth | True Positive 197 | False Negative 3 |
| Exp. No Growth | False Positive 62 | True Negative 12 |
| | Overall accuracy 76% | |

**C** **iML1515**

| | Predicted Growth | Predicted No Growth |
|---|---|---|
| Exp. Growth | True Positive 158 | False Negative 41 |
| Exp. No Growth | False Positive 28 | True Negative 46 |
| | Overall accuracy 75% | |

**D** **CANYUNs Model**

| | Predicted Growth | Predicted No Growth |
|---|---|---|
| Exp. Growth | True Positive 189 | False Negative 10 |
| Exp. No Growth | False Positive 13 | True Negative 61 |
| | Overall accuracy 92% | |

**Fig 5. The *E. coli* CANYUNs Model performs better than iML1515 and CarveMe when simulating growth on all known phenotypic data.** (A) The CarveMe model without gapfilling has a base accuracy of 52% and a Matthews Correlation Coefficient (MCC) of 0.09. (B) The CarveMe model we trained using all of the phenotypic data performs with an accuracy of 76% and an MCC of 0.29. However, there is a strong bias toward false positive predictions. (C) The manually curated *E. coli* K-12 model, iML1515, was not trained using all of the growth conditions. However, it performs with 75% accuracy and an MCC of 0.40 while maintaining a relatively even split between false positive predictions and false negative predictions. (D) The CANYUNs model we generated performs with 92% accuracy and an MCC of 0.78. The increased accuracy is primarily due to an improvement in true negative prediction rate.

condition. The most parsimonious solution is determined for each condition when using pFBA. In doing so, we are able to establish a more precise understanding of how the inclusion of genetic annotation evidence impacts the discovery of reactions when compared to the manually curated *E. coli* K-12 model, iML1515. We did not expect the CANYUNs reactions to align perfectly with iML1515 since the manual curation process did not include all of the growth conditions used to train the CANYUNs model. However, since the sequence-to-reaction dataset used to generate annotation evidence does not include all of the sequences used to build iML1515, we were able to track the FCRs that CANYUNs identifies without annotation evidence, yet are confirmed to be *E. coli* K-12 reactions by the iML1515 model. The 'Likely additions' category (Fig 6A and 6B) represents a set of reactions from the CANYUNs model with high genetic annotation evidence (bitscore above 500) that are not present in iML1515 and cannot be validated using this comparison, but they may represent reactions that could be added to iML1515 to improve alignment with the phenotypic data. We demonstrate that the dgFBA CANYUNs model has 12% greater alignment with iML1515 at the reaction level, compared to the pFBA CANYUNs model (Fig 6C). The discovery accuracy is calculated as the

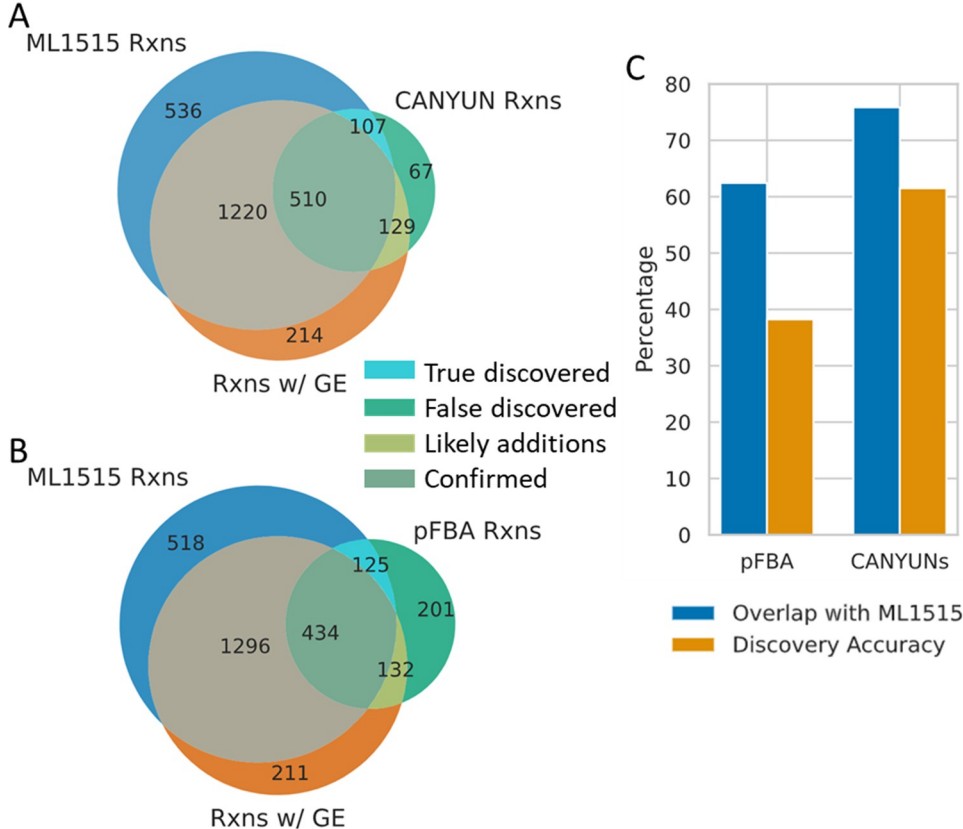

**Fig 6. CANYUNs reaction Certainty Values accurately identify reactions found in iML1515.** The manually curated metabolic network, iML1515, provides a point of comparison to determine if CANYUNs accurately identifies reactions for inclusion in the network. (A) By comparing the CANYUNs model with iML1515, we were able to place reactions into four categories. FCRs with genetic evidence and in iML1515 (confirmed), FCRs without genetic evidence in iML1515 (true discovered), FCRs with genetic evidence not in iML1515 (likely additions), and FCRs without genetic evidence and not in iML1515 (false discovered). The total amount of genetic evidence that is used to generate a CANYUNs model influences the accuracy of the FCRs. (B) When we use pFBA instead of dgFBA in the CANYUNs pipeline, there are far more reactions that lack genetic evidence and are not in iML1515. (C) The percent overlap of FCRs with reactions present in iML1515 increases from 62% when no genetic evidence is used (pFBA) to 76% overlap when all of the available genetic evidence is used.

number of FCRs that are identified by CANYUNs while lacking sufficient annotation evidence yet that were included in the iML1515 model. The dgFBA CANYUNs model has a discovery accuracy of 62%, 24% greater than the pFBA CANYUNs model. CANYUNs accurately identifies reactions that should be included in the *E. coli* K-12 metabolic network validated by the most recent manually curated reconstruction, iML1515.

A further analysis of the reaction CVs demonstrates that the accuracy of reaction inclusion in a CANYUNs model correlates positively with the magnitude of the reaction CV (Fig 7A and 7B). The percent overlap with iML1515 improves rapidly when the bottom 30 reactions with the lowest certainty values are ignored (Fig 7A and 7C). Overall, dgFBA provides a noticeable benefit over pFBA; however, there is a set of about 50 core reactions that are accurately identified with both optimization methods (Fig 7). We found that dgFBA strongly outperforms pFBA and CarveMe when evaluating the discovery accuracy. The performance of CANYUNs is in part explained by the reduced total number of discovered reactions compared to CarveMe. That number represents a significant advance when considering the process of manually validating the reactions with insufficient annotation evidence by searching for the appropriate gene-protein rule to add to the sequence-to-reaction dataset.

## CANYUNs model for the probiotic strain: *E. coli* Nissle

We built a novel model of the *E. coli* Nissle metabolic network to demonstrate the application of CANYUNs and to provide an example representation of a CANYUNs model with all accompanying source data. Nissle is a probiotic strain that has demonstrated measurable impacts on colonization resistance against human gastrointestinal pathogens [33–35]. Additionally, it is important to note that several studies demonstrate that the metabolism of Nissle is phenotypically different from K-12. We generated novel phenotypic growth data for *E. coli* Nissle using Biolog Phenotype MicroArray 96-well growth plates. We performed growth assays for the carbon source plates, PM1 and PM2A, in both aerobic and anaerobic growth conditions. We found that the metabolic consumption profile of Nissle is 9% different from K-12 (Fig 8A and S1 Table). There are 25 media conditions in which Nissle and K-12 do not align out of a total of 285. Nissle is able to grow in 16 conditions in which K-12 is not; in the other 9 conditions K-12 grows when Nissle does not. All inconclusive results for K-12 were treated as no growth conditions. Data for K-12 anaerobic growth in the PM2A plate does not exist on Ecocyc. All data is displayed in the Supporting Information (S1 and S2 Figs and S1 Table).

We generated a CANYUNs model using the same input biochemical network data discussed in Fig 1, while utilizing Nissle specific annotation evidence and our set of phenotypic growth experiments from culturing Nissle. The final model that we generated recapitulated the phenotypic growth data with an overall accuracy of 92%, with no false positive error (Fig 8B). There are 18 false negative conditions that could be fixed by manually adding organism-specific reactions to the model via expansion of the sequence-to-reaction dataset. There were 466 reactions that received CVs and had high genetic evidence, indicating that they are likely to be actively catalyzed by Nissle during exponential growth. Additionally, in the process of building the Nissle model, we identified 176 reactions that have low genetic evidence, yet were included in the model with CVs greater than zero (Fig 8C). These reactions represent those that may benefit from manual improvements to the sequence-to-reaction dataset to incorporate reference sequences that will better align with Nissle genes. There were an additional 5 spontaneous reactions included. The 1280 reactions that have genetic evidence in the Nissle genome, but are not included in the set of CANYUNs model reactions represent reactions that were not required for growth in any of the known growth conditions we utilized for building this model (Fig 8C). We assessed CVs associated with the 176 low evidence reactions to determine their rank of

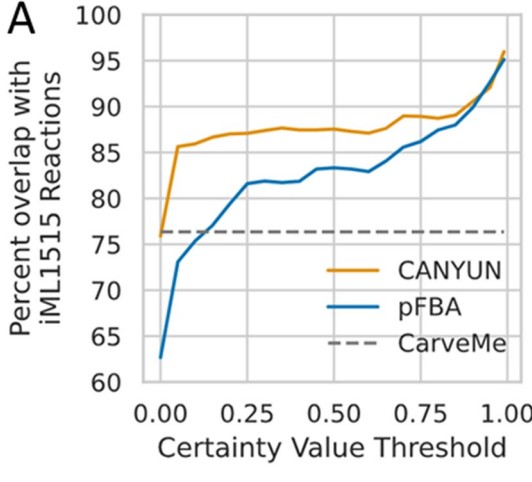

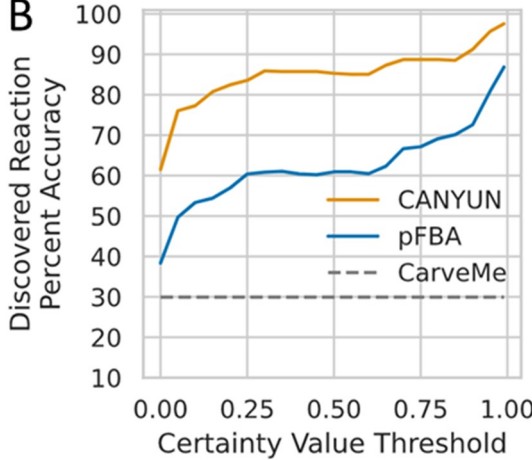

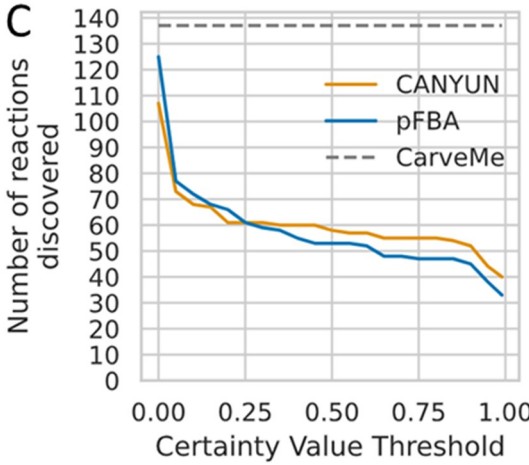

**Fig 7. Reaction Certainty Values correlate with accurate reaction inclusion and comparison with CarveMe.** (A) The percentage of reactions identified by CANYUNs that align with the iML1515 model correlates with the associated certainty value. All reactions with a certainty value greater than or equal to 0.99 have a 94% chance of being in the iML1515 model. (B) The accuracy of discovered reactions, with inclusion in iML1515 as reference, increases with the certainty values assigned using CANYUNs. (C) Although the accuracy of the discovered reactions increases with the certainty value, there is a significant drop in the number of reactions with the increase.

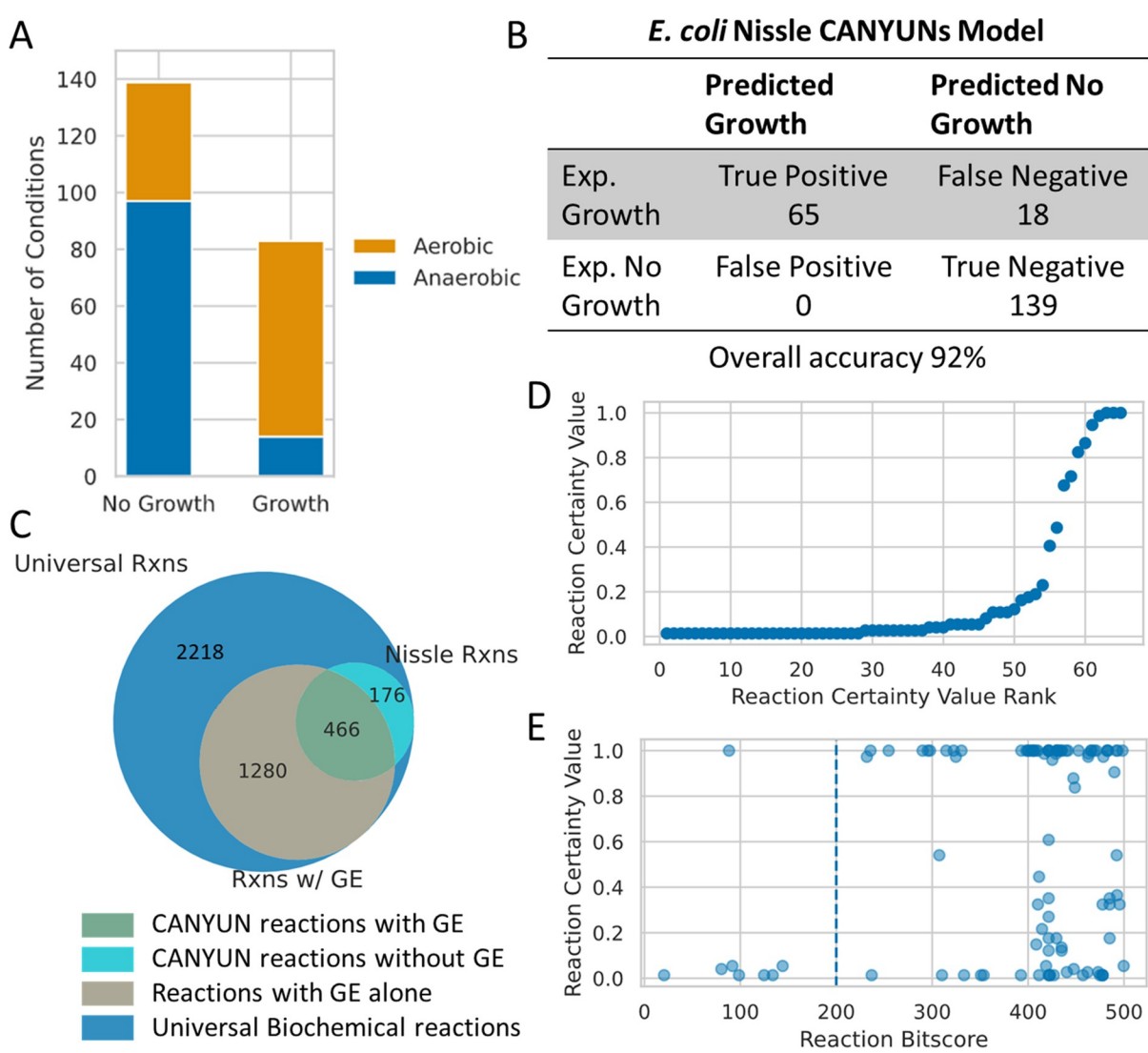

**Fig 8. *E. coli* Nissle Model.** (A) Phenotypic data used to build the model. (B) The final accuracy of the model is 92% with no false positive predictions. The model has an MCC of 0.83. (C) From the Universal Network, 1,746 reactions have Nissle specific genetic evidence associated with them; of those, there are 466 reactions that receive CVs from the CANYUNs pipeline. There are 176 reactions that do not have Nissle-specific genetic evidence, yet receive CVs and are thus included in the final CANYUNs model for Nissle. (D) These reactions received certainty values, but do not have associated genetic evidence; they are candidates for manually finding sequences to add to the reference dataset. (E) There is also a set of 103 reactions with CVs and low bitscores (below 500). Reactions with a high Certainty Value and a bitscore above 200 are likely candidates for direct additions to the sequence-to-reaction dataset.

importance for future curation of the datasets used to generate this model. There were 65 reactions within the set that have no reference sequences and thus have unknown genetic evidence (Fig 8D). Finally, there were 103 reactions that had both low genetic evidence and CVs greater than zero (Fig 8E). The reactions with bitscores closer to 500 and high CVs have high cumulative evidence indicating that they should be assessed further to determine the appropriate reference sequences that should be added to the sequence-to-reaction dataset used to generate this model.

## Discussion

Introduced here is a procedural metabolic network model construction method for the generation of CANYUNs models that accurately recapitulate phenotypic training data and select

appropriate reactions to represent the biochemical capabilities of a target organism. CANYUNs leverages a novel form of FBA, dgFBA, to direct flux through the universal metabolic network during model building and curation resulting in a GENRE that is structurally different from manually curated GENREs. Cumulative evidence for inclusion of a given reaction in a CANYUNs model is explicitly quantified during the network model construction process. Existing methods rely heavily on genetic data to estimate the metabolic capabilities of an organism. CANYUNs fills a separate niche; it produces procedurally generated GENREs that include functional data such as phenotypic growth data as an integral step in the curation protocol. Maintaining a strong connection with all source data allows CANYUNs to guard against information loss that can occur. Most importantly, a core objective of CANYUNs is to leverage the uncertainty innate to the biological data used during the reconstruction process to generate a GENRE built upon continuous data inputs. This aspect of CANYUNs models differs from the presence or absence of reactions in other GENREs. The structure of CANYUNs models allows for the uncertainty across biological data to be managed via redundancy. Each type of data provides various benefits while mitigating associated error.

With some exceptions, GENREs are frequently referred to in the literature as either a metabolic network reconstruction or a metabolic network model depending upon the context in which it is utilized. CANYUNs formalizes a structure that highlights the important differences between a network reconstruction and a model. Through this lens, a CANYUNs model can be viewed as a network reconstruction when including all of the evidence and data that is utilized (sequence-to-reaction dataset, universal metabolic network, phenotypic growth data, and the genetic evidence). Whereas, a CANYUNs model can be viewed and treated as a model when simulating flux through only the reactions that have non-zero CVs. The conceptual framework underlying this distinction is grounded in the idea that phenotypic growth data should be utilized to contextualize the genetic and biochemical data, rather than determine the absolute inclusion or exclusion of reactions from a GENRE. By accepting that an organism-specific GENRE is simply a contextualized version of the underlying universal metabolic network, there is additional flexibility that can be leveraged for future curation of the GENRE with additional biological data or expansion of the source data. This concept is a core difference between CANYUNs and other methods for generating metabolic models or GENREs.

The technical characteristics that make CANYUNs unique from other methods revolve around a consistent connection to source data and management of associated uncertainty in the source data. CANYUNs models are structured to facilitate future curation by ensuring that all source data is an integral part of the model. As seen in Fig 8C, a CANYUNs model when viewed as a GENRE consists of four classifications of reactions: Universal biochemical reactions, reactions with genetic evidence and no CV (Certainty Value), reactions with genetic evidence and a CV, and reactions with a CV and no genetic evidence. Each class of reaction has an associated continuous spectrum that indicates how much evidence has contributed to the reaction being in that class. Universal biochemical reactions have a spectrum of reference sequences (Fig 1D). Reactions with only genetic evidence have the reaction bitscore which is positioned on a continuous spectrum. Reactions with only a CV have the magnitude of the CV that represents the cumulative phenotypic and biochemical evidence associated with the reaction. Finally, reactions with a CV and genetic evidence have the most complex array of associated evidence including: genetic, phenotypic, and biochemical.

Procedural generation methods benefit from existing manually curated GENREs via their contribution to the universal biochemical network and the associated sequence-to-reaction reference dataset. Manually curated versions of foundational data provide the base on which procedural generation methods can be built upon. It has been shown that procedurally generated GENREs benefit from manually curated data inputs [14]. For example, ensuring that all

reactions in the universal biochemical network are mass-balanced and that there are no mass-generating loops in the network reduces the need for additional curation of the resulting GEN-REs [14,36]. The high specificity required for the annotation of metabolic enzymatic function with accompanying thermodynamic and stoichiometric directionality is relatively unique to GENREs and thus a limiting factor in the building process. Reaction directionality is a simple, yet important aspect of curating GENREs. Often sequence annotation databases do not include specific information about reaction directionality. Directionality can become particularly important when a reaction is thermodynamically unfavorable in a certain direction. Improper directionality assignments can lead to free mass-generation and improper assignment of catalytic function. The CANYUNs method provides a way to quantify reaction evidence specific to directionality by calculating CVs specific to the direction of the flux through reactions. This level of specificity provides more control over the behavior of an organism-specific network.

Genetic data is the base on which GENREs are built, yet not all genetic information is required to represent the metabolic network for an organism. Due to gene regulation and other aspects of metabolic control that are exceptionally challenging to incorporate into a GENRE, it is important to keep in mind that genetic data, with all associated uncertainties, is simply an imperfect lens through which an organism-specific model can begin to take shape. Functional phenotypic data, when paired with a stoichiometrically accurate universal metabolic network, provides information for contextualizing the underlying genetic data. This conceptual framework provides the flexibility required for passively allowing unknown gene regulation across differing growth conditions to influence the building process. The core assumption in this conceptual model is that thermodynamic efficiency, both stoichiometric and enzymatic, is the primary governing objective at the cellular level. This objective is technically achieved during the prediction of growth by utilizing only the reactions with CVs for flux balance analysis. All other reactions with genetic evidence alone did not demonstrate their activity during the training of the model and are thus not active for growth predictions.

CANYUNs has several limitations that are important to understand. The set of reactions included in a CANYUNs model only includes the reactions that receive CVs as a result of utilizing dgFBA to simulate growth in each known growth condition. Therefore, the resulting network is best thought of as a contextualized model based on all of the known growth conditions. A result of this model building process is a small network of reactions that have high genetic evidence and allow for high accuracy when recapitulating the phenotypic data utilized for building the model. Given the nature of a CANYUNs model, it is not designed to make predictions about growth in unknown conditions, rather it is designed to provide an accurate representation of the reactions that carry flux in the known growth conditions. Another limitation that currently exists is the use of binary growth data, CANYUNs would benefit from expanding the pipeline to incorporate exponential growth rates associated with each growth condition. Finally, CANYUNs relies on BLAST, a slow alignment method, for generating the reaction bitscores. In order to apply CANYUNs to much larger datasets, a more efficient alignment tool would be required.

A core focus of this study was the need for better curation of source data used for procedurally generated GENREs. Curation of these datasets is far more useful for future model generation, opposed to the curation of specific models. The curation of specific models, separate from the source datasets, can result in thermodynamic inconsistences among models that make it difficult to simulate metabolic interactions. This method lays the groundwork for data-driven expansion of the sequence-to-reaction dataset by quantifying phenotypic evidence for the addition of sequences that are slightly below the functional bitscore threshold (set to 500 in this study) to the sequence-to-reaction dataset. Thus, phenotypic data could be utilized through CANYUNs to systematically expand the sequence-to-reaction dataset and help to

annotate other genomes. With enough phenotypic data from an array of organisms, it would be possible to conservatively expand the reference dataset to propagate well-defined functional annotations to many more sequences and thus expand the ability to generate accurate models across a wide array of organisms. Additionally, CANYUNs models optimize for false negative predictions, thus specifically identifying areas of the universal biochemical network that require the manual addition of reactions. CANYUNs provides solutions for several challenges in expanding genome-scale metabolic network reconstructions to model the vast array of microbes that exist in human-associated microbial communities.

## Methods

### Universal metabolic network curation

We started with the CarveMe universal model and added any new reactions from iML1515 to make the universal metabolic network. Any metabolites with multiple formulas were altered to maintain only the metabolite formula used in the iML1515 model. Metabolites with multiple formulas that are not in the iML1515 model were adjusted based on stoichiometric consistency across all reactions. The final universal metabolic network is available on Github.

We utilized the following optimization problem to determine if the universal metabolic network contained any thermodynamically infeasible mass generation. The problem maximizes flux through a set of sink reactions that allow flux to leave the system from within the cellular or periplasm compartment. No metabolites are allowed to enter the system through exchange reactions. This basic formulation of FBA provides an output of all reactions that are able to carry flux when no external metabolites are provided. The simultaneous maximization of flux through all sink reactions allows for a thorough evaluation of all possible mass generating loops.

*Optimization problem* for *free mass generation check*

*Maximize* : $\sum \overrightarrow{v_{snk}}$

*subject to*

$$S \cdot \vec{v} = 0$$
$$\overline{LB} \leq \vec{v} \leq \overline{UB}$$
$$\vec{0} \leq \overrightarrow{v_{snk}} \leq \overrightarrow{1000}$$
$$\vec{0} \leq \overrightarrow{v_{ex}} \leq \vec{0}$$

$\overrightarrow{v_{snk}} = Sink\ rxn\ flux\ vector$

$S = stoichiometric\ matrix$

$\vec{v} = $ intracellular flux vector

$\overline{LB} = lower\ bound\ vector$

$\overline{UB} = upper\ bound\ vector$

$\overrightarrow{v_{ex}} = exchange\ rxn\ flux\ vector$

### Data guided flux balance analysis

Data Guided Flux Balance Analysis is formulated in Fig 2. In the presented formulation the total sum of weighted flux through all reactions is maximized. The reactions are weighted based on the associated genetic evidence. Reactions with greater evidence have a higher reaction weight and are thus more likely to carry flux in the final solution. All reversible reactions are represented as two opposing reactions and the net flux is calculated to determine the flux value associated with the reversible reaction. This formulation of dgFBA is a linear

optimization problem, unlike the similar problem presented in the CarveMe publication which is a mixed integer linear program (MILP) [14]. Although not the driving reason for developing dgFBA, the fact that it is an LP allows for the solution to theoretically be solved faster than a MILP with the same number of variables. A key advantage of dgFBA is that it allows for the calculation of certainty values that can then be utilized in the rest of the CANYUNs pipeline to quantify cumulative evidence for the inclusion of reactions in an organism specific model. However, an important limitation of dgFBA is that the resulting flux distribution is not representative of actual biological flux that may be present in a given growth condition. A separate FBA problem must be solved in order to calculate a flux distribution that can be compared to experimentally-measured biological flux in a metabolic network.

## CANYUNs model building process

We utilized the sequence-to-reaction database provided in the CarveMe publication. We aligned the unknown protein sequence fasta file with Diamond to calculate sequence alignment bitscores for each protein [37]. We then calculated reaction bitscores for each reaction in the universal biochemical network utilizing the CarveMe method. However, we did not normalize the bitscores, as done previously [14]. For the CANYUNs model construction, the superset of all flux-carrying reactions determined using dgFBA for each of the known growth conditions are utilized, while excluding all other reactions from the model. A condition was considered to have simulated growth when the flux through the biomass reactions, using FBA, resulted in a flux greater than 0.1. Reversible reactions are represented as two opposite reactions that can only proceed in the forward direction; the sum of their opposing flux is then calculated to find the net flux through the reversible reaction. We then calculate the ratio of times that each reaction carries flux across all growth conditions to determine the certainty values for each reaction. Each reaction receives two certainty values, one for each direction, the sum of these two values is never greater than 1. We utilize all reactions with non-zero certainty values to build a draft CANYUNs model.

The draft model is processed to determine all of the conditionally essential reactions for each of the draft model growth predictions (true and false positives). The reactions that are conditionally essential for more false positives than true positives are reactions that can be used to improve predictive accuracy. The reaction with the most leverage to improve model predictions is removed from the CANYUNs model to create the final model. All reaction bounds were set to zero, negative 1000, or positive 1000 for the CANYUNs model building process. Reactions that never carried flux in the known growth conditions were removed from the final CANYUNs model. The model building process requires roughly 30 minutes from start to finish on an Intel Xeon processor with 4 cores. The resulting CANYUNs model was assessed using MEMOTE to determine that it has 100% Stoichiometric Consistency.

## CarveMe model generation

We utilized CarveMe in a Windows 10 command line to generate a base model without gapfilling and a gapfilled model with all known growth conditions. All default parameters were used [14]. The phenotypic growth data for *E. coli* K-12 was acquired from the EcoCyc database (https://biocyc.org/ECOLI/NEW-IMAGE?object=Growth-Media). These data are also available on GitHub.

## *E. coli* Nissle data collection and model generation

We cultured *E. coli* Nissle in Biolog plates and evaluated growth with a TECAN microplate reader. Optical density measurements were performed using a 600-nanometer wavelength. We used Biolog PM1 and PM2A plates. The cultures were started with an overnight culture in

M9 4% glucose medium at 37 degrees Celsius from a single colony selection off an LB agar plate. The cells were centrifuged and washed with PBS three times and finally resuspended and diluted into the base Biolog inoculation fluid. The resulting OD of the Biolog inoculation fluid, after dilution, was calculated to be 0.01 OD. When cultured anaerobically the OD of the plates were measured at the end of 40 hours shaking in an anaerobic chamber. The baseline OD for each well was determined by filling a plate with the base media alone. The OD of the aerobic plates was measured every 10 minutes for the whole time-course with shaking.

We acquired the Nissle genome from EMBL. With this genome we implemented CANYUNs to calculate CVs and build an organism-specific GENRE. The resulting CANYUNs model was assessed using MEMOTE to determine that it has 100% Stoichiometric Consistency.

## Supporting information

**S1 Fig.** *E. coli* **Nissle PM1 Biolog Growth Data.** There are 70 aerobic growth conditions and 34 anaerobic growth conditions. The x-axis is time in hours and the y-axis is OD measured at 600 nanometers.
(TIF)

**S2 Fig.** *E. coli* **Nissle PM2A Biolog Growth Data.** There are 22 aerobic growth conditions and 9 anaerobic growth conditions. The x-axis is time in hours and the y-axis is OD measured at 600 nanometers.
(TIF)

**S1 Table. Difference in** *E. coli* **Nissle growth compared to** *E. coli* **K12.** Anaerobic data for K-12 growth in Biolog plate PM2A was not available on EcoCyc.
(DOCX)

## Author Contributions

**Conceptualization:** Thomas J. Moutinho, Jr., Jason A. Papin.

**Data curation:** Thomas J. Moutinho, Jr., Benjamin C. Neubert.

**Formal analysis:** Thomas J. Moutinho, Jr.

**Funding acquisition:** Thomas J. Moutinho, Jr., Matthew L. Jenior, Jason A. Papin.

**Investigation:** Thomas J. Moutinho, Jr.

**Methodology:** Thomas J. Moutinho, Jr., Benjamin C. Neubert, Matthew L. Jenior, Jason A. Papin.

**Project administration:** Thomas J. Moutinho, Jr.

**Resources:** Jason A. Papin.

**Software:** Thomas J. Moutinho, Jr., Benjamin C. Neubert, Matthew L. Jenior.

**Supervision:** Jason A. Papin.

**Validation:** Thomas J. Moutinho, Jr., Benjamin C. Neubert.

**Visualization:** Thomas J. Moutinho, Jr., Matthew L. Jenior, Jason A. Papin.

**Writing – original draft:** Thomas J. Moutinho, Jr.

**Writing – review & editing:** Thomas J. Moutinho, Jr., Benjamin C. Neubert, Matthew L. Jenior, Jason A. Papin.

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
