## [Decision Letter · Decision Letter 0]

17 Sep 2021

Dear Jason,

Thank you very much for submitting your manuscript "Quantifying cumulative phenotypic and genomic evidence for procedural generation of metabolic network reconstructions" for consideration at PLOS Computational Biology.

As with all papers reviewed by the journal, your manuscript was reviewed by members of the editorial board and by several independent reviewers. In light of the reviews (below this email), we would like to invite the resubmission of a significantly-revised version that takes into account the reviewers' comments.

We cannot make any decision about publication until we have seen the revised manuscript and your response to the reviewers' comments. Your revised manuscript is also likely to be sent to reviewers for further evaluation.

Sincerely,

Kiran Raosaheb Patil, Ph.D.

Deputy Editor

PLOS Computational Biology

Reviewer's Responses to Questions

**Comments to the Authors:**

Reviewer #1: In the present paper Thomas J. Moutinho Jr. and colleagues presented a novel method for quantifying the combined genomic, biochemical, and phenotypic evidence for all the biochemical reaction during automated genome-scale metabolic network reconstruction. Authors reported that the method generates accurate GENREs supported by a quantitative metric describing the cumulative evidence for each biochemical reaction included in the network. Additionally, they built a new CANYUN GENRE for E. coli Nissle using phenotypic data specifically collected for the project.

The paper is clear and well-written, the cited literature covers in general the previously published studies related to the topic, despite the addition of some other references was suggested (see details below).

Detailed comments to the authors.

General comment. The approach described here can be useful to improve the annotation process in general and not only when propedeutic to the generation of a GENRE. Authors could consider to include a general comment suggesting similar improvements for the software used for microbial genome annotation (here annotation refers to functional assignment).

General comment. Authors tested their approach in a microbial species (E. coli) which is well studied, easy to annotate and easy to grow. I am wondering if the present approach can also be applied to other microbial species which are taxonomically distant from other well studied microbes and difficult to cultivate. Obviously, limitations in growing microbes cause limitations in obtaining phenotipic data. Authors can provide a short explanation regarding the limitations of the proposed approach.

General comment. Consistency is important to ensure the theoretical soundness of models (e.g., mass and charge balance, absence of energy-generating cycles) and model validation is important to ensure that a model describes the biological reality of the organism it is representing. Consistency should be systematically curated and assessed, preferably passing the standardized MEMOTE test suite (https://memote.io/).

General comment. The github page of the tool is not weel structured, a readme and a manual page are missing.

General comment. To date 16/08/2021 the _init.py_ file in Github is empty; probably this is ok, but please verify.

General comment. The data utilized for validating CANYUN on E. coli K-12 are only briefly mentioned and should be described more in detail with related web links for clarity and full reproducibility.

General comment. The methods should provide more details over the simulation parameters, such as metabolite exchange bounds, non-null growth threshold, etc. to allow full reproducibility.

General comment. More precise inputs, data requirements, and limitations should be provided for a more accessible utilization by other researchers. For example, while it is clear that CANYUN was developed and validated with binary growth/no-growth phenotypic screens, a potential user might wonder if other phenotypic data could be used. If only binary growth/no-growth datasets are supported by CANYUN, this should be clearly discussed along with its other limitations.

Line 7. Authors can consider to include recently reported collections of GENREs such as those obtained in (doi: 10.1038/s41559-020-01353-4) and in (https://doi.org/10.1016/j.ymben.2020.08.013).

Line 65. Please consider to include more references such as (doi: 10.1093/bioinformatics/btab324).

Line 99. The authors selected CarveMe as a tool to generate the draft GENRE; did authors consider other tools as well? (e.g. RAVEN, gapseq, Merlin)? Why did authors specifically selected CarveMe?

Line 110; lines 151-156. Authors used BLASTp to align the genome of the target organism with reference sequences in the CarveMe sequence-to-reaction dataset. The sequence alignment bitscores for E. coli K-12 genes and the CarveMe dataset were used to generate reaction bitscores using previously published methods. However, the simple use of BLASTp can limit the efficiency of the process, did authors consider the possibility to use other tools such as for example HMM search?

Lines 421 and following. The generation of models for poorly studied microorganisms is not explained exhaustively, in particular the importance of obtaining biochemical data and to perform growth tests should be clarified.

Line 470 and following. Information about the solver(s) used has not been provided and should be included in Materials and Methods.

Lines 328 and following. This last part is not very easy to understand. I would expect to see some results reporting at what level the implementation of the proposed approach and the generation of the CANYUN GENRE can recapitulate in a more reliable way the results obtained from biolog experiments.

Reviewer #2: Moutinho and colleagues describe a new computational framework for automatic generation of genome-scale reconstruction networks that improves upon pre-existing methods by integrating genomic, biochemical, and phenotypic (i.e. growth conditions) to more accurately represent organism-specific metabolism by constraint-based modeling. There is a pressing need in genome-scale metabolic modeling to eliminate manual curation of reconstructed networks due the sheer number of bacterial species and strains implicated in microbiome applications; thus the impact of this paper is potentially high. I particularly like how the proposed model curation methodology leverages the wealth of phenotypic data available to add confidence in a metabolic network based upon ensemble performance simulating the experimental conditions. The authors compare “ground truth” of a manually curated model to a model generated by their pipeline and evaluate accuracy with pre-existing computational tools as a means of benchmarking their results. They test their computational tools on a new E. coli Nissle network by generating new experimental Biolog growth data for the purposes of this study.

I find only minor issues with the manuscript in its current form which, if addressed, will increase the readership and usage of their CANYUNs code.

1. Understandably the authors are focused on human health applications with the K-12 strain, however the CANYUNs applications could be easily extended to environmental microbiota (water sources, soil samples) and the paper could be generalized slightly in the introduction to extend potential adoption of the pipeline.

2. Along the same lines, there is no associated documentation in the github README file for actual implementation of the files deposited. More description for the user here would be helpful.

3. The description of calculated reaction bitscores, which is central to the generation of the CANYUNs, confidence in reactions, and data guided FBA interpretation of the models, seems incomplete. The authors cite reference 10 on the bitscore calculation. The original CARVEME algorithm yields a log-normal distribution in scoring. The distribution of reaction bitscores used to define thresholds in Figures 1 and 2 do not display this feature, and there is not discussion on the implications of this for the weights applied in the dgFBA generation.

4. The Certainty Value is a useful construct for quantifying confidence in a reaction by the frequency of flux occurring across all experimental conditions, thereby leveraging numerous media growth condition datasets. While this may be defined as a simple ratio, it should be explicitly included in the CANYUNs pipeline description in the Methods section.

Reviewer #3: The manuscript proposes a new framework for generating genome-scale metabolic network reconstructions. Instead of making a draft metabolic network and subsequently performing a gap-filling, the authors propose to simultaneously integrate genome annotation and phenotypic growth data. The authors demonstrate through a well-chosen set of comparative studies the usefulness of such a holistic approach.

While the manuscript is well-written and the text is, for the most part, clear, some methodological passages need additional clarification and discussion. My specific comments follow.

Comments:

1. A similar optimization problem of maximizing the number of high genomic evidence reaction and minimizing low evidence reactions already exists in CarveMe. Though the authors are apparently aware of this work, they do not discuss what are the advantages and disadvantages of the LP dgFBA formulation compared to the MILP CarveMe formulation. In the discussion, it would be important to place both optimization problems in the context of their enveloping methods (CANYUN and CarveMe). The readers should be able to assess clearly the contribution of this work and this discussion would be a key part in this process.

2. Whereas this might be obvious to some of the readers, the elements in the reaction flux vector, v, in dgFBA can take only positive values (otherwise, the reactions in the forward direction with the high genetic annotation evidence and the reactions in the reverse direction with the low evidence would have the same weight in the objective). The authors should state this explicitly in their problem formulation (Fig. 2) to avoid confusion.

In this line, since the directionality of the reactions directly influences the criterion function, one has to explore all possible combinations of forward/reverse reactions to obtain the optimal objective value. Considering that in the studied network there are 45 reactions that can operate in either of the directions (line 221), it seems that one has to analyze 2^45 cases, which is computationally intractable. One can reduce this number by performing a flux coupling analysis and assign certain directionalities from the literature, however, the number of the cases can still be daunting.

It seems that this issue of the proposed optimization problem is completely neglected in the manuscript. This issue is not new and specific only to this work and it hampers some other methods in the field of constraint-based modeling, however, it has to be acknowledged and discussed how to alleviate it.

3. Lines 212-214, the authors state “the directionality of each flux value is used to specifically determine the cumulative evidence for each reaction specific to direction”. The authors did not specify precisely how this is done. In the light of my remark 2, when testing the cumulative evidence of one reaction, the chosen directionality of other reactions will influence the results. This should be discussed and clarified.

4. Though dgFBA identifies a set of reactions that can carry flux given certain phenotypic data, I am not sure that the authors can talk about “flux distributions” of dgFBA in the sense that dgFBA solutions would probably not have flux values corresponding to the actual intracellular fluxes. For example, in pFBA, the solutions would likely correlate to the metabolic states with minimal dissipation. In contrast, in dgFBA, the reactions with weights close to zero can take as high values as the steady-state constraint (S*v=0) allows.

5. The authors curate universal network by manually removing reactions that were contributing to free-mass generation. Once the proposed method identified the reactions taking part in the free-mass generation, it is not clearly explained whether all reactions that are taking part in the free-mass generation are removed or just a few of them. If only a part of reactions is removed, how the authors choose which reactions to remove and what are the recommendations to do this? Otherwise, if all reactions are removed, why the authors emphasize that they are removed “manually”?

Further down in the results section, how many reactions were removed? Were the removed reactions originating from iML1515, the CarveME network, or both? Moreover, it is reasonable to assume that these reactions, if appearing in iML1515 or in the created CarveME model would affect the essentiality analysis. Can you comment on this?

6. In general, for testing this and other possible issues in the genome-scale models, the MEMOTE method by C. Lieven et al (Nat Biotech 38, pp 272–276 (2020)) is the community standard. Were the obtained models tested using this tool? Would MEMOTE identify the issues with the free-mass generation and if yes, how the proposed method is different than the one used in MEMOTE?

7. Concerning the confusion matrices presented in Figure 5, the authors use as the performance measure the accuracy. Considering that the analyzed sets are not symmetric, a more balanced measure for this would be the Matthews correlation coefficient.

8. Lines 389-391, the authors state that there is no need for further thermodynamic curation of the resulting GENREs – thermodynamics can also give us information about thermodynamically-feasible directionality of reactions in the network, which can reduce the number of reactions operating in both forward and reverse direction.

9. It would be good that the authors discuss why the number of false negatives from 3 in the initial CANYUN model (Fig. 4d) increased to 10 in the final model (Fig. 5d).

10. Figure 1, it seems that the network has more of the exchange metabolites than the exchange reactions. Can you explain that?

In the caption for the panel e), the authors should mention what is the orange dashed line and explain the bar left to that line (1/3 of reactions with zero bitscore).

11. It is not entirely clear to me what was the purpose of showing the distribution of reaction weights in the forward and reverse direction (Figure 2c and d)? Besides, these figures were not referred to in the main manuscript.

12. Figure 4a, b, and, c what is the information that the authors wanted to convey and how to read these panels?

13. Line 459, cite the Diamond method.

**Have the authors made all data and (if applicable) computational code underlying the findings in their manuscript fully available?**

Reviewer #1: Yes

Reviewer #2: Yes

Reviewer #3: Yes

PLOS authors have the option to publish the peer review history of their article (what does this mean?). If published, this will include your full peer review and any attached files.

Reviewer #1: No

Reviewer #2: No

Reviewer #3: No
---

## [Decision Letter · Decision Letter 1]

19 Jan 2022

Dear Professor Papin,

We are pleased to inform you that your manuscript 'Quantifying cumulative phenotypic and genomic evidence for procedural generation of metabolic network reconstructions' has been provisionally accepted for publication in PLOS Computational Biology.

Best regards,

Kiran Raosaheb Patil, Ph.D.

Deputy Editor

PLOS Computational Biology

Reviewer's Responses to Questions

**Comments to the Authors:**

Reviewer #1: After checking the revised version of the manuscript 'Quantifying cumulative phenotypic and genomic evidence for procedural generation of metabolic network reconstructions' sumitted by Thomas J. Moutinho Jr. and colleagues, I can state that it has been substantially improved. Authors implemented, when possible, the modifications requested by the reviewers.

According to my evaluation, the manuscript, in its present form, is suitable for publication in PLOS computational biology.

Reviewer #3: The authors have addressed my concerns and I recommend the publication of this manuscript.

**Have the authors made all data and (if applicable) computational code underlying the findings in their manuscript fully available?**

Reviewer #1: Yes

Reviewer #3: Yes

PLOS authors have the option to publish the peer review history of their article (what does this mean?). If published, this will include your full peer review and any attached files.

Reviewer #1: No

Reviewer #3: No

---

## [Editor Report · Acceptance letter]

2 Feb 2022

PCOMPBIOL-D-21-01397R1 

Quantifying cumulative phenotypic and genomic evidence for procedural generation of metabolic network reconstructions

Dear Dr Papin,

I am pleased to inform you that your manuscript has been formally accepted for publication in PLOS Computational Biology. Your manuscript is now with our production department and you will be notified of the publication date in due course.

With kind regards,

Agnes Pap
